# Incidence and Characteristics of Intraocular Lens Dislocation after Phacoemulsification: An Eight-Year, Nationwide, Population-Based Study

**DOI:** 10.3390/jcm10173830

**Published:** 2021-08-26

**Authors:** Ga-In Lee, Dong Hui Lim, Sang Ah Chi, Seon Woo Kim, Jisang Han, Dong Wook Shin, Tae-Young Chung

**Affiliations:** 1Department of Ophthalmology, Samsung Medical Center, Sungkyunkwan University School of Medicine, Seoul 06351, Korea; porishi71@gmail.com (G.-I.L.); tychung@skku.edu (T.-Y.C.); 2Samsung Advanced Institute for Health Sciences & Technology, Sungkyunkwan University, Seoul 06355, Korea; 3Department of Health Sciences and Technology, SAIHST, Sungkyunkwan University, Seoul 06355, Korea; sszzgg91@naver.com; 4Statistics and Data Center, Research Institute for Future Medicine, Samsung Medical Center, Seoul 06351, Korea; 5Biostatic and Clinical Epidemiology Center, Samsung Medical Center, Seoul 06351, Korea; seonwoo.kim@samsung.com; 6Department of Ophthalmology, Sungkyunkwan University School of Medicine, Kangbuk Samsung Hospital, Seoul 03181, Korea; jhanoprs@gmail.com; 7Supportive Care Center, Samsung Comprehensive Cancer Center, Samsung Medical Center, Seoul 06351, Korea; dongwook.shin@samsung.com; 8Department of Family Medicine, Samsung Medical Center, Sungkyunkwan University School of Medicine, Seoul 06351, Korea

**Keywords:** IOL dislocation, incidence, national data

## Abstract

Background: We investigate the incidence and characteristics of IOL dislocation among the pseudophakic population after phacoemulsification. Methods: National data were collected from the health claims recorded with the Health Insurance Review and Assessment Service of South Korea from 2009 to 2016. Pseudophakic patients aged 40 years or older were included. The incidence estimates of phacoemulsification and IOL dislocation were analyzed, and the cumulative probabilities of IOL dislocation among the pseudophakic population and general population were calculated as a proportion. Results: Of 51,307,821 total subjects, 25,271,917 of whom were aged 40 years or older, 3,906,071 cataract cases in 2,650,104 pseudophakic patients were identified, and 72,309 patients experienced IOL dislocation. The cumulative probability was 2.73% per person and 1.85% per surgery among patients 40 years of age or older. The eight-year incidence rate for IOL dislocation in the pseudophakic population aged 40 years or older was 7671 per 1,000,000 person-years (95% CI: 7616–7727), including 10,341 cases in men and 5814 in women. Incidence peaked in the seventh decade of life for cataract surgery but in the fifth decade of life for IOL dislocation. The cumulative probability of IOL dislocation after phacoemulsification was approximately 2%, and the incidence rate was about 7000 per 1,000,000 pseudophakic patients. Conclusions: There was a significantly higher incidence of IOL dislocation among young males, even though the higher incidence of cataract surgery was observed among older females. These estimates of the nationwide, population-based incidence of IOL dislocation can help increase understanding of the population vulnerable to IOL dislocation.

## 1. Introduction

The pseudophakic population has experienced tremendous growth as the aging population continues to expand and life expectancy increases. The number of surgeries has increased exponentially, while remarkable improvements in cataract surgery have occurred in the form of phacoemulsification with ultrasonography with femtosecond laser-assisted device operation. Consequently, the number of complications following cataract surgery is also growing. A severe complication that may present is intraocular lens (IOL) dislocation. Since the first case was reported in the 1970s [1], there have been several publications involving cases of IOL dislocation [2,3,4,5]. IOL dislocation usually requires secondary surgical management and could possibly lead to other complications such as retinal tears, retinal detachment, vitreous hemorrhage, or secondary glaucoma. As the pseudophakic population continues to increase in size due to the development of cataract extraction devices and extended lifespans, secondary surgical procedures after cataract surgery may rapidly emerge as a social issue and become more of a burden to the public health care.

The incidence rate of late-onset IOL dislocation varies among worldwide studies from 0.05% to 1.7% [3,6,7,8,9]. Previous investigations have employed various study designs and numbers of cases. Further, incidence rates are different between countries, with a United States (US) cohort study showing a similar incidence rate over a 30-year study period [7] but a Swedish study showing a stably higher incidence rate because of the higher prevalence of pseudoexfoliation syndrome in this latter group [8,10]. Based on different proportions of IOL dislocation among countries, it could be meaningful to evaluate the incidence of IOL dislocation among Asians. Few studies have reported the incidence of IOL dislocation in Asia.

South Korea has ideal conditions for an epidemiological study due to the mandatory universal health insurance system, which captures the medical care data of the entire national population of 50 million people. We, therefore, conducted the present study using data from a large population cohort in South Korea. The same authors have previously published an article on the risk factor of IOL dislocation using the same nationwide big registry data [11]. The aim of our study was to investigate the incidence of cataract surgery and IOL dislocation according to age and sex in the general population.

## 2. Methods

### 2.1. Data Source

This was a nationwide, population-based study performed using data from the Korean national health claims database of patients with IOL dislocation and secondary IOL implantation. We accessed health claims recorded by the Health Insurance Review and Assessment (HIRA) service of South Korea from 2009 to 2016. This database holds all health care utilization information for both inpatients and outpatients using codes from the Korean Standard Classification of Diseases, seventh revision, with a few changes to the Korean situation based on the International Classification of Diseases, 10th revision. This database has been used previously to describe epidemiologic features and analyze risk factors of other diseases with detailed information [12,13,14]. The HIRA reviews all health claims in South Korea, including those submitted through the Korean National Health Insurance scheme, which covers 97% of the South Korean population. Other available medical assistance programs (e.g., the Medical Assistance Program and Medical Care for Patriots and Veterans Affairs Scheme) cover the remaining 3% of the population. Accessible data include diagnoses, date of visit, procedures, socioeconomic level, prescription records, comorbidities, and demographic features of almost the entire population. Patients in the HIRA database can be identified easily by their unique Korean resident registration number; this guarantees the absence of duplication and prevents loss of data. To define the health insurance-covered population of South Korea as a denominator, annual data from the Health Insurance Statistics (Korean Statistical Information Service; available at http://kosis.kr, accessed on 29 August 2018) were used. The HIRA Deliberative Committee approved the use of the HIRA database for data collected from 2009 through 2016 for this study. The study was performed following the principles of the Declaration of Helsinki and approved by the Samsung Medical Center Institutional Review Board.

### 2.2. Participants and Definition

The pseudophakic population was defined as (1) patients with intraocular lens (registration code Z961) or (2) evidence of cataract surgery by phacoemulsification (registration code S5119) and primary IOL implantation (registration code S5117) on the same day. As we described in detail previously [11], to mitigate the risk of coding errors in the database, we adjunctively assessed the codes of implanted IOLs. Additionally, cataract surgery is covered by the national diagnosis-related groups (DRGs) insurance system in Korea, which provides a flat payment determined by the government based on several factors such as diagnosis, severity, and the procedures performed. The hospital administrates the diseases or procedures covered by the DRG system strictly; thus, coding errors are rare in cataract-related diagnostic and procedure codes. For incidence estimates of cataract surgery, we used the health insurance-covered population during the study period as the denominator. The population used to evaluate the incidence rate was determined after removing the identified cases that underwent cataract surgery each year from the Population and Housing Census.

Afterward, relevant cases were defined as those including diagnosis of IOL dislocation (registration codes H271 and T852) who underwent secondary IOL implantation (registration codes S5116 and S5118) during the eight-year study period (2009–2016). Patients with diagnostic codes only were included in the IOL dislocation population as mild cases; however, patients with surgical codes were required also to have a diagnostic code to exclude secondary IOL cases with other causes such as aphakia, anisometropia, refractive errors, dysphotopsia, and IOL opacity. For incidence estimates, the date of the earliest claim with registration codes of S5119 and S5117, followed by H271, T852, S5116, or S5118 was defined as the index date and considered the incident time, with the patient being considered an incident case in that year. We did not include IOL dislocation cases on the same day or one day after the cataract surgery, because the etiology or natural history in these very early cases is very different. For instance, intraoperative posterior capsular tears can cause immediate IOL decentration or subluxation. Patients with a congenital disease such as Marfan syndrome, congenital cataract, and/or congenital anomaly were excluded. All patients with the aforementioned registration codes had confirmed diagnoses and were registered by an ophthalmologist.

### 2.3. Statistical Analyses

For incidence estimates of cataract surgery, we calculated the number of patients or cases who underwent cataract surgery divided by person-time of the total population. The person-time incidence rates of IOL dislocation for 2009 to 2016 were calculated as the number of people in whom IOL dislocation developed and who underwent a secondary surgical procedure divided by total person-time at risk during the study period. Therefore, person-years were counted after the incident time. The cumulative probabilities of IOL dislocation among pseudophakic and general populations were calculated as a proportion. Additionally, the male–female ratios for incidence rates were analyzed. The process of identifying IOL dislocation is presented in Figure 1. Poisson distribution was used to estimate the 95% confidence intervals (CIs) of the incidence rates. We used the SAS Enterprise Guide version 9.3 software program (SAS Institute, Cary, NC, USA) for all analyses.

## 3. Results

### 3.1. Demographics, Cumulative Probability, and Incidence

Among 25,271,917 subjects, 3,906,071 cataract cases in 2,650,104 pseudophakic patients were identified, with a mean follow-up of 7.48 years. Of these patients, 1,067,243 (40.27%) were male, and 1,582,861 (59.73%) were female. The mean age of the pseudophakic patients in the age group of 40 years of age or older was 68.72 ± 9.85 years, while 72,309 patients experienced IOL dislocation with a mean follow-up of 7.56 years, comprising 39,989 (55.30%) men and 32,320 (44.70%) women. The mean age of patients with IOL dislocation who were 40 years of age or older was 66.53 years (median age: 67 years). The cumulative probability of IOL dislocation was slightly decreased to 2.73% per person and 1.85% per surgery among this pseudophakic population.

### 3.2. Incidence of Cataract Surgery and IOL Dislocation

The eight-year incidence estimate of cataract surgery in the general population aged 40 years or older was 10,774 per 1,000,000 person-years (95% CI: 10,760–10,788), comprising 9121 men and 12,318 women. Among the general population, the incidence rate for IOL dislocation for the eight years of the study was 360 per 1,000,000 person-years (95% CI: 350–370), comprising 413 men and 311 women. Finally, the incidence rate for IOL dislocation for the eight years of the study in the pseudophakic population aged 40 years or older was 7671 per 1,000,000 person-years (CI: 7616–7727), comprising 10,341 men and 5814 women (Table 1). 

### 3.3. Age and Sex Predominance

The peak age of incidence was 75 to 79 years among all pseudophakic patients, 75 to 79 years in males, and 70 to 74 years in females. The male–female overall ratio of cataract surgery was 0.74, with a female predominance. Notably, the peak age of incidence for IOL dislocation in the general population was 75 to 79 years, whereas the peak age in the total pseudophakic population was 40 to 44 years, 50 to 54 years in males, and 40 to 44 years in females (Figure 2). The male–female overall ratio of IOL dislocation was 1.78, with a male predominance.

## 4. Discussion

### 4.1. Incidence of Cataract Surgery

In a previous US study, the incidence of cataract surgery substantially increased at the age range of 25 to 32 years [15,16]. Klein et al. also reported as part of the Beaver Dam Eye Study that the incidence of lens extraction has increased over the past 20 years in people older than 65 years [17]. Separately, the incidence rate for eight years of cataract surgery in this study decreased slightly from 2012 to 2013 and then tended to increase again according to the development of cataract surgery techniques and changes in the national insurance system with the addition of DRGs since 2012 (Appendix A). In the DRG system, health care providers seek efficiency to receive more rewards for achieving consistent outcomes and low costs of treatment because a flat payment is charged regardless of the difficulty of the cataract surgery [18]. This may have influenced the flow of change in this study. Moreover, it was confirmed that females more frequently underwent cataract surgery than males in this study, as has been similarly reported in other studies in Korea and the US Medicare population [18,19]. Park et al. additionally reported a higher incidence of cataract surgery among women based on other national data [20]. Since women have a longer lifespan and a higher prevalence of cataracts than men, the rate of cataract surgery in women is higher than that in men.

### 4.2. Incidence of IOL Dislocation

In previous studies of incidence estimates, there has been no consensus on whether IOL dislocation is increasing in prevalence (Table 2). Furthermore, there have been no data published on Asian subjects specifically. As iris color and structure are different between Western and Asian populations, the results of anterior chamber inflammation, capsular problems, and zonular weakness after cataract surgery may also be dissimilar [21]. For example, pseudoexfoliation is a well-known risk factor of IOL dislocation among the Caucasian population. However, our previous study reported that preoperative pseudoexfoliation was not a significant risk factor of IOL dislocation [11]. The prevalence of pseudoexfoliation shows large regional variation, and the reported prevalence in Asia was 0.4% to 3.4% lower than that in Western countries [22,23,24,25,26]. In this study, the number of incidences differs from a previously published study of risk factors of IOL dislocation. The reason is that the previously published study only targeted patients who have actually undergone cataract surgery since 2002 in order to more strictly evaluate the risk factor of IOL dislocation; however, this study is to inspect the incidence of IOL dislocation. All the pseudophakic population who had undergone cataract surgery before 2002 were included as an incident case or target population at risk; therefore, the more target subjects became the at-risk group—eventually, the number was increased from 15,170 to 72,309.

In a Swedish population, Monestam et al. reported that 5 (0.6%) of the total 800 patients at risk experienced a late IOL dislocation. However, this study found that 521 of 810 patients were lost to follow-up during the 10-year period; therefore, the incidence in this cohort is likely to be underestimated [10]. Jakobsson et al. reported a hospital-based case series with 84 cases, where the incidence of surgery for late dislocated IOL was 0.042% to 0.052% in the pseudophakic population and 0.027% to 0.033% in pseudophakic eyes during 2004 to 2006 [6].

Dabrowska-Kloda et al. used regional cohort data of approximately 550,000 patients and reported 140 cases of IOL dislocation requiring surgical intervention with a 10-year cumulative probability of 0.55% and a 20-year cumulative probability of 1.0% [8]. The incidence of dislocation increased by approximately twofold with an almost threefold increase in pseudophakic patients over 20 years. The data of these studies were estimated from the Swedish National Cataract Register because of an inability to obtain accurate at-risk population data regarding the number of pseudophakic eyes. In a US cohort study, the 10-year cumulative risk of IOL dislocation was 0.1% to 1.7%, maintaining a similar trend after 25 years [7]. The Swedish authors explained that a higher prevalence of pseudoexfoliation, which is a significant risk factor of IOL dislocation in the Scandinavian population, might be related to the higher incidence of IOL dislocation after cataract surgery, as reported in a Norwegian study [27].

Both studies, however, showed a trend of increasing incidence of IOL dislocation and suggested the increase in cataract surgery as the reason. However, in our study, the characteristics of patients with cataract surgery and IOL dislocation showed a different pattern. Among the general population, the incidence rates of both cataract surgery and IOL dislocation peaked at 75 to 59 years of age, but male sex was predominant among cases of IOL dislocation in contrast to a female dominancy in cataract surgery. If the population was limited to pseudophakic patients, the incidence rate of IOL dislocation showed a peak age at 40 to 44 years with a male predominance. Importantly, the increase in the pseudophakic population is not the direct reason for the increasing IOL dislocation. Pseudoexfoliation increases with age, and a lower prevalence of pseudoexfoliation in Asians than in Western people may affect the results. Pseudoexfoliation is associated with zonular weakness and is a known risk factor of IOL dislocation. Traumatic cataract cases, another well-known risk factor of IOL dislocation and much more common in young males than older and/or females, might affect the higher incidence of IOL dislocation in young males.

Recently, Bothun et al. reported long-term follow-up results in which 80 cases of IOL dislocation were detected among 148,201 regional cohort patients, with an incidence of 28.4 per million person-years and a three-year cumulative probability of 1.5% [9]. However, we do not know whether all 148,201 patients were pseudophakic. People who have not undergone cataract surgery were included in the study as denominators to calculate the person-years. In 2007, the incidence of cataract surgery was reported in the same cohort; therefore we can assume that the pseudophakic population may be identified by the data [9,15]. This is the reason for the large difference in incidence rate between studies, as mentioned above.

Another difference of note is the definition of IOL dislocation. We included every IOL dislocation case with or without secondary surgery; therefore, mild cases were also included. All diagnoses were performed by an ophthalmologist, as even a small degree of IOL dislocation can progress and cause severe vision problems. This approach is more helpful clinically to report the actual incidence of IOL dislocation.

Additionally, we postulated that increasing IOL dislocation might be the result of changes in the technique of cataract surgery (i.e., phacoemulsification). Compared with extracapsular cataract extraction (ECCE), phacoemulsification is more likely to produce zonular stress. An Australian population study reported that IOL dislocations have almost doubled in prevalence from 0.17% in 1985 to 1989 to 0.3% in 1995 to 2001, although this is still much lower than that in the current study [28]. This could be affected by a larger proportion of the study participants being treated with ECCE (36.8%) in comparison with recent years. After the late 2000s, phacoemulsification has evolved into the technique of choice in most cataract surgeries [15]. In light of this situation, the pre-2000 cohort population that underwent cataract surgery with ECCE is not appropriately comparable.

### 4.3. Age-Specific Incidence Rates of IOL Dislocation

A particular point about the age-specific incidence rates among the general population is that the peak age of IOL dislocation was 70 to 74 years, which was similar to in a Swedish nested case–control study reporting an age of 73.1 years [8]. However, among the pseudophakic population in our study, the highest rates were in the age range of 50 to 54 years among males and 40 to 44 years among females. This means that there was a significant proportion of participants who underwent cataract surgery in old age. This leads to a considerable number of person-years. Pershing et al. reported that procedure-related revisits or readmissions following cataract surgery were highest for patients in their 20s and 30s and lowest for patients in their 70s, respectively [29]. The authors explained that younger patients are more likely to have other causes of earlier cataracts (e.g., prior trauma, other eye diseases, or surgery) leading to riskier and more complicated surgery.

### 4.4. Sex-Specific Incidence Rates of IOL Dislocation

Another interesting point of the sex-specific prevalence and incidence rates is that males had a higher rate than did females at all ages, despite the larger population of women who underwent cataract surgery [15,16,30]. We speculated that males tend to more often participate in rough activities than females. The male sex could thus be more vulnerable to trauma, which is a risk factor of IOL dislocation. Our study found a higher male–female ratio in younger ages (40s–50s) than in older ages (>50s). Pershing et al. reported that male sex is a predisposing factor associated with readmission following cataract surgery, similar to our findings [29].

In this study, cataract surgery was more common in women and older ages, whereas IOL dislocation was found more often in men and younger ages. Therefore, an increase in the number of cataract surgeries may not necessarily have an impact on IOL dislocation. The study used recent data with homogeneous characteristics, although they were from a relatively short-term period and may be limited in impact since the number of cataract surgeries has not dramatically increased from 2009 to 2016. Accordingly, this bias can be ruled out, and young age and male sex can be ruled as significant factors of IOL dislocation.

Our study has several advantages that must be highlighted. First, we made every effort possible to avoid selection bias and observed the distributions of age and sex using national data for all citizens with a large number of participants based on a clear diagnosis established by an ophthalmologist. Second, this study aimed for applicability to the entire Korean population as the first Asian study of its kind. Previous Caucasian population-based investigations in Olmsted (124,000), Värmland county (280,000), and the Rochester Epidemiology Project (115,000), which included relatively isolated semi-urban counties, are difficult to apply to Asian patients. Additionally, there may be differences in the structure of Asian eyes such as zonules and pseudoexfoliation in comparison with Caucasians. There are quite a few studies or case series on analysis of IOL dislocation focusing on risk factors in Asia [4,31,32]. We targeted the entire Korean population (*n* = 50,000,000) and established representativeness for analysis of incidence. Third, the incidence was accurately determined by defining both the entire population and a distinct pseudophakic population. Finally, it was a distinctive population that included almost all the people of a country who underwent the same surgical method of phacoemulsification. The results are therefore representative, reliable, and valuable for comparison with previous outcomes.

There are also several limitations to this study. We included IOL dislocation based on both diagnostic and surgical codes, as confirmed by an ophthalmologist, but very mild dislocation cases without surgery could have been excluded if ophthalmologists did not enter the correct diagnostic code for IOL dislocation; consequently, our incidence could be underestimated. However, these mild decentration or subluxation cases do not affect visual outcomes and require no treatment; therefore, they should not be counted. The study duration was relatively short, at eight years, and therefore, IOL dislocation occurring later than eight years was not detected, which could cause underestimation of incidence. We could not access the HIRA data for other years. Due to this limitation, patients with cataract surgery, followed by secondary IOL dislocation registered before 2009, could not be excluded because we were not allowed to access the HIRA database for earlier years. Lastly, in the previous study, we investigated the risk factors of IOL dislocation with the HIRA database [11]. The preoperative history of ocular comorbidities such as uveitis (hazard ratio, HR 1.59), pseudoexfoliation (HR 2.31), traumatic cataract (HR 6.48), brunescent cataract (HR 1.83) significantly increased the risk of IOL dislocation. The postoperative ophthalmic variables including partial vitrectomy (HR 12.48), iris and ciliary body injuries referred to as loose zonule (HR 7.18), glaucoma (HR 3.12), and retinal surgery (HR 3.79) also significantly increased the risk of IOL dislocation. However, lack of clinical information for confounders such as demographics, comorbidities, refractive error, visual acuity, type of IOL material, intraoperative problems, the experience level of the surgeon, and phacoemulsification time could be another limitation of the present study. Further studies of the risk factors of IOL dislocation are warranted.

In conclusion, this is an important research question in terms of public health and policy. In addition, a paucity of data exists on IOL dislocation in the Asian population, and this study addresses this issue. The cumulative probability of IOL dislocation after phacoemulsification was 2.73% per person, and the incidence rate for eight years was 7671 per one million. The incidence rates of IOL dislocation in males and females peak in the early 50s and early 40s, respectively, and there were higher incidence rates in males than in females at all ages despite the tendency of women to undergo more frequent cataract surgery. These estimates of the nationwide, population-based incidence of IOL dislocation might be valuable information for consideration in further study of the phenomenon in Asians.

## Figures and Tables

**Figure 1 jcm-10-03830-f001:**
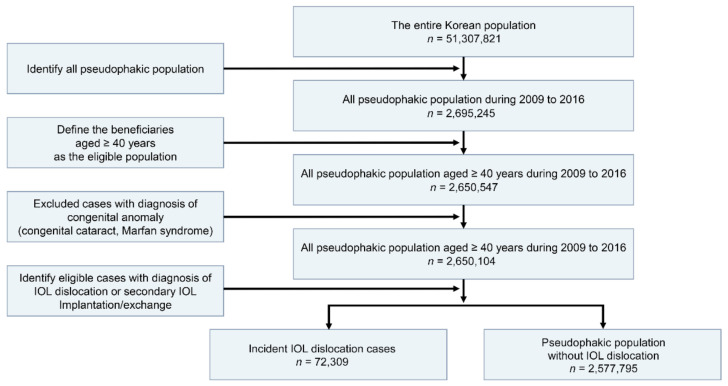
The process of identifying IOL dislocation cases.

**Figure 2 jcm-10-03830-f002:**
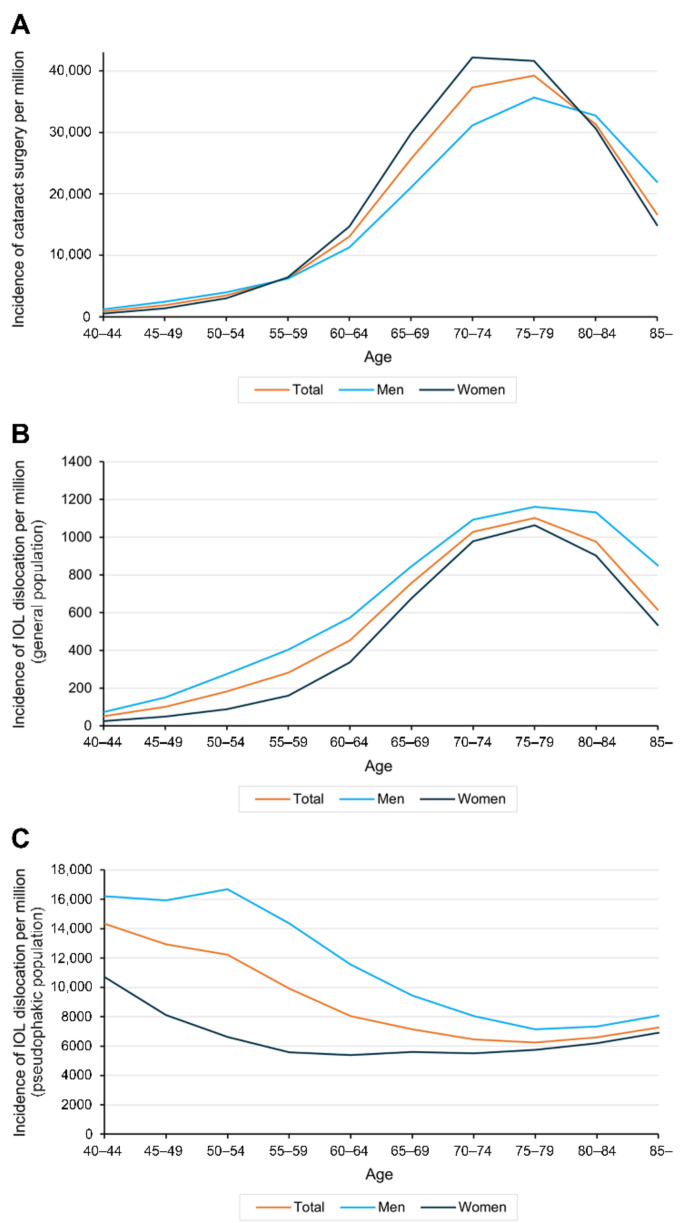
The incidence rate of cataract surgery and IOL dislocation with age distribution: (**A**) the peak age of incidence was 75 to 79 years in all pseudophakic patients, 75 to 79 years in males, and 70 to 74 years in females; (**B**) the peak age of incidence for IOL dislocation in the general population was 75 to 79 years, (**C**) whereas the peak age in the pseudophakic population was 40 to 44 years, including 50 to 54 years in males and 40 to 44 years in females.

**Table 1 jcm-10-03830-t001:** The number of patients with IOL dislocation and estimated cumulative incidence rate (per 1,000,000 person-years) of IOL dislocation in the South Korean pseudophakic population during 2009–2016.

Age Group (yrs)	Total			Male			Female			M to F Ratio
	No.	Incidence ^a^	95% CI	No.	Incidence ^a^	95% CI	No.	Incidence ^a^	95% CI	
40–44	1833	14,338	13,681–14,994	1368	16,209	15,350–17,068	465	10,703	9730–11,676	1.51
45–49	3555	12,993	12,508–13,358	2698	15,936	15,335–16,538	857	8116	7573–8660	1.96
50–54	6227	12,220	11,917–12,524	4729	16,687	16,211–17,163	1498	6623	6288–6958	2.52
55–59	7888	9911	9692–10,129	5632	14,371	13,996–14,747	2256	5584	5353–5814	2.57
60–64	9146	8057	7892–8223	5681	11,563	11,262–11,863	3465	5382	5203–5561	2.15
65–69	11,967	7136	7009–7264	6336	9439	9206–9671	5631	5600	5453–5746	1.69
70–74	13,699	6462	6354–6571	6419	8044	7847–8240	7280	5508	5381–5634	1.46
75–79	10,478	6250	6130–6369	4349	7144	6932–7356	6129	5740	5596–5884	1.24
80-84	5334	6588	6411–6764	2014	7344	7023–7665	3320	6200	5989–6411	1.18
85+	2182	7271	6966–7576	763	8063	7491–8635	1419	6907	6547–7266	1.17
Overall	72,309	7671	7615–7727	39,989	10,340	10,239–10,442	32,320	5814	5750–5877	1.78

No. = number of individuals; CI = confidence interval. ^a^ Per 1,000,000 person-years.

**Table 2 jcm-10-03830-t002:** Previously reported incidence rates of IOL dislocation in different studies.

Authors	Country	Publication Date	Cohort Population	Eligible Pseudophakic Population	Number of IOL Dislocation Cases	Studied Duration	Design	Source	Incidence Rate	Cumulative Probability
(1) Monestam et al. [10]	Sweden	2009	ND	239 survivors among 810 participants	5	10 years	Retrospective Cohort	Hospital	-	10-year 0.6% (5/800)
(2) Jakobsson et al. [6]	Sweden	2010	-	-	84	3 years	Case series	Hospital	-	Estimated as 0.05% per person 0.03% per surgery
(3) Pueringer et al. [7]	United States	2011	124,277	14,471 cases in 9577 residents	16	30 years (1980-2009)	Retrospective cohort. Nested case–control	Rochester Epidemiology Project	-	5-year 0.1% 10-year 0.1% 15-year 0.2% 20-year 0.7% 25-year 1.7%
(4) Dabrowska-Kloda et al. [8]	Sweden	2015	ND	approximately 550,000 ^a^	140	20 years (1992–2012)	Retrospective cohort. Nested case–control	Regional (Varmland County). Hospital	-	5-year 0.09% 10-year 0.55% 15-year 1.0% 20-year 1.0%
(5) Bothun et al. [9]	United States	2018	148,201	ND	80	30 years (1986–2016)	Retrospective cohort	Rochester Epidemiology Project	28.4 per 1,000,000 person-year	1.5% for 30 years
Present study	South Korea	-	25,271,917	3,906,071 cases in 2,650,104 people	72,309	8 years (2009–2016)	Retrospective cohort	Nationwide Registry data	360/general 7.671/pseudophakia per 1,000,000 person-year	2.73% per person 1.85% per surgery for 8 years

^a^ The data were estimated from the Swedish National Cataract Register, the Department of Ophthalmology of the County Hospital of Varmland.

## Data Availability

All datasets generated and/or analyzed during the current study are available from the corresponding author upon reasonable request.

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
