# Peer review of "Incidence and Characteristics of Intraocular Lens Dislocation after Phacoemulsification: An Eight-Year, Nationwide, Population-Based Study"

_jcm, 2021, doi:10.3390/jcm10173830_

Round 1

Reviewer 1 Report

Lee et al present a well structured and designed research paper on the incidence of IOL dislocation after uncomplicated phacomeulisification. The sample size is dramatically high with a great impact on final results.

I have some doubts on the real impact of these data. Undoubdetely, it is fundamental to investigate the incidence of cataract surgery and IOL dislocation acording to age and sex in general population.

However, I believe that, more than age or sex, it is fundamental to investigate other factors of IOL dislocation such as:

  1. causes (pseudoexofliatio lentis, trauma, inflammatory eye disease, high myopia, loose zonula)
  2. Type of IOL implanted (single piece, 3 pieces, ecc...)
  3. Type of prmary surgery (complications?)

Sex and age are fundamental factors but unchangeable. On the contrary the previous mentioned factors allows the surgeon to apply strategy to prevent dislocation (such as capsular tension ring implantatio, or 3 piecese IOL).

I would suggest, if it possible, to include also these data in the analysis, becasue with a such large sample size results could be dramatically importantat. This wil make the study of great scientific relevance and application.

Author Response

Thank you for your suggestion. In previous study (reference #11 in the manuscript), we investigated the risk factors of IOL dislocation. In the multivariate analysis adjusted for age and sex, preoperative history of uveitis (hazard ratio, HR 1.59), pseudoexfoliation (HR 2.31), traumatic cataract (HR 6.48), brunescent cataract (HR 1.83) significantly increased the risk of IOL dislocation. The postoperative ophthalmic variables such as partial vitrectomy (HR 12.48), iris and ciliary body injuries referred as loose zonule (HR 7.18), uveitis (HR 2.52), glaucoma surgery (HR 3.12), and retinal surgery (HR 3.79) also significantly increased the risk of IOL dislocation. However, this national data was limited by a lack of specific clinical information, such as refractive error, type of IOL material, use of capsular tension ring, intraoperative problems, and phacoemulsification times, all of which may have been potential confounders. It seems to be of great research if a prospective study including various factors that can be changed is conducted in the future.

  1. Lee, G.I.; Lim, D.H.; Chi, S.A.; Kim, S.W.; Shin, D.W.; Chung, T.Y. Risk Factors for Intraocular Lens Dislocation After Phacoemulsification: A Nationwide Population-Based Cohort Study. American journal of ophthalmology 2020, 214, 86-96.

Reviewer 2 Report

Vering interesting and large registry based study. Why didn't the authors calculate correlations with

  • indication for cataract surgery
  • grade of cataract
  • ocular comorbidities such as pseudoexfoliation etc.
  • IOL type / design   (the authors stated that the codes for the IOLs have been recorded)
  • IOL implantation site (capsular bag, sulcus, other)

There is some confusion with Figure 2: B and C cannot be both the incidence of IOL dislocation per million.

Any ideas why the incidence of dislocation in men is larger than in women? Should be discussed. 

It would be very interesting to see what amount of dislocation has been used to define "dislocation"? Is it more than 1 mm of decentration or complete dislocation from the implantation site?

Author Response

Vering interesting and large registry based study. Why didn't the authors calculate correlations with indication for cataract surgery

grade of cataract

ocular comorbidities such as pseudoexfoliation etc.

IOL type / design (the authors stated that the codes for the IOLs have been recorded)

IOL implantation site (capsular bag, sulcus, other)

RESPONSE: Thank you for your suggestion. In previous study (reference #11 in the manuscript), we investigated the risk factors of IOL dislocation. In the multivariate analysis adjusted for age and sex, preoperative history of uveitis (hazard ratio, HR 1.59), pseudoexfoliation (HR 2.31), traumatic cataract (HR 6.48), brunescent cataract (HR 1.83) significantly increased the risk of IOL dislocation. The postoperative ophthalmic variables such as partial vitrectomy (HR 12.48), iris and ciliary body injuries referred as loose zonule (HR 7.18), uveitis (HR 2.52), glaucoma surgery (HR 3.12), and retinal surgery (HR 3.79) also significantly increased the risk of IOL dislocation. However, this national data was limited by a lack of specific clinical information, such as grade of cataract, visual acuity, refractive error, type of IOL material, use of capsular tension ring, intraoperative problems, and phacoemulsification times, IOL implantation sites, all of which may have been potential confounders. It seems to be of great research if a prospective study including various factors that can be changed is conducted in the future.

  1. Lee, G.I.; Lim, D.H.; Chi, S.A.; Kim, S.W.; Shin, D.W.; Chung, T.Y. Risk Factors for Intraocular Lens Dislocation After Phacoemulsification: A Nationwide Population-Based Cohort Study. American journal of ophthalmology 2020, 214, 86-96.

There is some confusion with Figure 2: B and C cannot be both the incidence of IOL dislocation per million.

RESPONSE: Thank you for your comment. Sorry for the confusion. Please refer to the Figure legend.

Figure 2. The incidence rate of cataract surgery and IOL dislocation with age distribution.

(A) The peak age of incidence was 75 to 79 years in all pseudophakic patients, 75 to 79 years in males, and 70 to 74 years in females. (B) The peak age of incidence for IOL dislocation in the general population was 75 to 79 years, (C) whereas the peak age in the pseudophakic population was 40 to 44 years, including 50 to 54 years in males and 40 to 44 years in females.

Figure 2B indicates incidence of IOL dislocation based on pseudophakic population and Figure 2C indicates incidence of IOL dislocation based on health insurance-covered registered population of South Korea. We changed the vertical column of Figure 2.

Any ideas why the incidence of dislocation in men is larger than in women? Should be discussed.

RESPONSE: Thank you for your suggestion. We already discussed in sub-section “sex-specific incidence rates of IOL dislocation” (Page 14 line 329-344). It is hypothesized that males tend to more often participate in rough activities than females. The male sex could be more vulnerable to trauma, which is a risk factor of IOL dislocation. Likewise, a study using national data in Japan demonstrated that late recurrent IOL dislocation occurred frequently in younger men than in women.1 Also, Pershing et al. showed that male sex was a predisposing factor associated with readmission after cataract surgery.2

<References>

  1. Kawano, M. Takeuchi, S. Tanaka, T. Yamashita, T. Sakamoto, K. Kawakami. Current status of late and recurrent intraocular lens dislocation: analysis of real-world data in Japan. Jpn J Ophthalmol, 63 (1) (2019), pp. 65-72
  2. Pershing, D.E. Morrison, T. Hernandez-Boussard. Cataract surgery complications and revisit rates among three states. Am J Ophthalmol, 171 (2016), pp. 130-138

It would be very interesting to see what amount of dislocation has been used to define "dislocation"? Is it more than 1 mm of decentration or complete dislocation from the implantation site?

RESPONSE: Our study included IOL dislocation cases based on both diagnostic and surgical codes. Thus, very mild dislocation cases that have no effect on vision and only require observation have been excluded if ophthalmologists did not enter the diagnostic codes. We defined IOL dislocation as clinically meaningful rather than the amount of IOL dislocation.

Round 2

Reviewer 1 Report

No other comments

Reviewer 2 Report

-